# CHU9D Normative Data in Peruvian Adolescents

**DOI:** 10.3390/jpm11121272

**Published:** 2021-12-02

**Authors:** Roxana Paola Palacios-Cartagena, Raquel Pastor-Cisneros, Jose Carmelo Adsuar, Jorge Pérez-Gómez, Miguel Ángel García-Gordillo, María Mendoza-Muñoz

**Affiliations:** 1Promoting a Healthy Society Research Group (PHeSO), Faculty of Sport Sciences, University of Extremadura, 10003 Cáceres, Spain; rppalacioscartagena@gmail.com (R.P.P.-C.); jadssal@unex.es (J.C.A.); mamendozam@unex.es (M.M.-M.); 2Health, Economy, Motricity and Education Research Group (HEME), Faculty of Sport Sciences, University of Extremadura, 10003 Cáceres, Spain; jorgepg100@unex.es; 3Facultad de Administración y Negocios, Universidad Autónoma de Chile, Sede Talca 3467987, Chile; miguelgarciagordillo@gmail.com

**Keywords:** adolescents, CHU9D, health conditions, health-related quality of life (HRQOL), health status assessment

## Abstract

Background: Due to the vulnerability to protective and risk factors during adolescence, there is a growing interest in the study of health-related quality of life (HRQoL) at this stage. The CHU9D is a generic and practical HRQoL instrument that provides values on all dimensions of self-perceived health, in addition to providing utilities and a cost-utility assessment fee, unlike other instruments. This study was conducted with an adolescent population in Peru. The main objective of this article is to report the normative values of the CHU9D questionnaire in Peruvian adolescents. Methods: The CHU9D questionnaire was administered to Peruvian adolescent students. A total of 1229 young people participated in the survey. The CHU9D score was reflected as a function of gender, age, weight, height, and educational level. Results: The mean CHU9D utility index for the total sample was 0.890; this rating was significantly better for boys with 0.887 and girls with 0.867. The ceiling effect was higher for male adolescents with 32.6 than for female adolescents. Conclusions: The results of the present study show that adolescents in school show a positive perception of HRQoL. It is also concluded that the CHU9D instrument can be effectively applied to economic evaluations for interventions to improve the quality of life of adolescents.

## 1. Introduction

Adolescence is a critical period, when biological, psychological, and social changes occur [1]. At the biological level, various hormonal changes occur in the body such as acne and, in the case of women, the onset of menstruation. At the psychological level, they present identity crises, emotional alterations, unawareness to face risks, search for autonomy, are defiant and tend to generate conflict situations. In the social sphere, they renounce their dependence on their parents, they distance themselves from their family, and their interest in friendships grows [2]. These changes may affect the adolescent’s HRQoL.

Health-related quality of life (HRQoL) can be defined as the intrinsic feeling, conditioned by the current state of one’s health. Initially, this concept referred to the physical, psychological, and social aspects, but over time, it has shifted toward aspects of well-being, happiness, and activity in other areas that are fundamental to the person [3]. In relation to HRQoL from an adolescent’s perspective, it is an essential contribution as this is a stage vulnerable to protective factors and risk factors. At this stage, they do not manifest serious illnesses unlike other age groups, but face difficult circumstances that affect their mood [4].

To assess HRQoL there are generic and specific tools such as self-report questionnaires in children and adolescents. Among the most widely used are: the EuroQol [5], the CHQ (Child Health and Illness Profile) [6], KIDSCREEN-27 [7], the PedsQL [8], and the Child Health Utility-9 D (CHU9D) [9]. The latter is a practical and simple to apply instrument, which provides values across all dimensions of self-perceived health. In addition, it has utilities and a tariff that can be used to assess cost-utility, which is an advantage over other instruments.

The CHU9D comes from the UK, culturally adapted into English, Canadian, Dutch, Swedish, Danish, Italian, Welsh, Portuguese, Japanese, French, and Spanish (Measuring and Valuing Health 2021). It is an instrument that was originally developed for use with children and young people. Internationally, there is growing interest in the application of this questionnaire as numerous scientific studies have demonstrated its validity and reliability [9]. This measure has been demonstrated in a variety of medical conditions in which to confirm the effectiveness of the tool in specific areas and monitoring the progression of patients (children and adolescents) through the disease. It can therefore be seen as a brief and easy to complete instrument as it does not involve specific diseases [10].

CHU9D has also been used in a pilot clinical study with children and adolescents in a diabetic population [11]. It has also been used in a dental research study with children in a New Zealand population [12]. It is also used in research programs, evaluation, and treatment options for type 1 diabetes [13] as well as in mental health [14] and obesity prevention [15]. Similarly, in a study conducted in Australia, where the instrument was applied with young adolescents, the results showed that the CHU9D is a reliable and valid instrument for this population group [10].

Having normative data makes it possible to contrast the HRQoL of the general and pathological population, providing information on the differences between them, matching the characteristics of the subjects as well as age- and sex-specific conditions. In this way, it contributes to the evolution and planning of health policies. Thus, the aim of this study was to describe the perceived quality of life in Peruvian adolescent students.

## 2. Materials and Methods

### 2.1. Study Design

A single-measure cross-sectional study was conducted.

### 2.2. Ethics Approval

Ethical approval was granted by the bioethics and biosafety committee of the University of Extremadura on 10 December 2020 (approval number: 162/2020), in accordance with the updates of the Declaration of Helsinki, as amended by the 64th General Assembly of the World Medical Association (Fortaleza, Brazil, 2013) and the Law 14/2007 on Biomedical Research.

### 2.3. Sample Calculation

A random sample of 1153 individuals is sufficient to estimate, with 95% confidence and a precision of +/–2.5 percentage units, a population percentage that is expected to be around 25% [16]. All this takes as a reference the total population of Peru (33,353,304 inhabitants) in 2020 according to the National Institute of Statistics [17].

### 2.4. Participants

Data collection took place in school or after-school sports activities. Via mobile phone, students accessed the survey link by completing the questionnaire. All participants met the following inclusion criteria: (1) being between 12 and 17 years of age; (2) informed consent signed by parents or legal guardians; and (3) acceptance of the participant in the study.

#### Sample Size

The total sample consisted of 1229 adolescent students of which 622 were female (50.6%) and 607 male (49.4%), aged 12–17 years with a mean age of 14.62 (±1.64).

### 2.5. Procedures and Measures

The socio-demographic characteristics collected in the survey were: age, sex, weight, height, and educational level. 

The CHU9D is a generic preference-based HRQoL measure developed specifically for use with young people. Previous studies have validated the CHU9D for use in older adolescent populations aged 11–17 years [10,18]. This self-report instrument, which aims to measure adolescents’ self-perceived HRQoL, is composed of nine items, each with five response categories (scored from 1 to 5). It assesses the adolescents’ day-to-day functioning in the following domains: worry, sadness, pain, tiredness, discomfort, school, sleep, daily routine, and activities [19].

### 2.6. Statistical Analyses

All information collected was tabulated in a database designed specifically for this study. Statistical analyses for this study were conducted using IBM SPSS Statistics software (version 25, IBM, Chicago, IL, USA) and the participants’ personal data were kept anonymous.

Data are represented by mean and standard deviation, and median and interquartile range for the total sample, divided by sex, age, and body weight category.

The internal reliability of the scale was defined by the Cronbach’s coefficient of the total scale. 

The discriminative ability of the questionnaire was analyzed using the area under the ROC curve (AUC). An area of 1 represents perfect classification, while an area of 0.5 represents an absence of classification accuracy. ROC values of >0.90 are considered excellent, 0.80–0.89 good, 0.70–0.79 acceptable, and <0.70 poor [20].

The Mann–Whitney U test was used to analyze sex differences. The chi-square test was used for multiple comparisons of categorical variables (age and body weight categories).

The ceiling effect was calculated as the percentage of participants who obtained the best possible health status (111111111).

The Child Health Utility Index 9D (CHU9D Utility Index) was calculated using the existing algorithm in the UK adult general population developed by Stevens (2012).

## 3. Results

To analyze the discriminative ability of the questionnaire, the area under the ROC curve (AUC) of healthy and cancer participants was analyzed. The area under the ROC curve was 0.86, with a sensitivity and specificity of 100.0 and 71.03, respectively, for a cut-off point of 0.71. Furthermore, with respect to the reliability of the scale, the Cronbach’s α for the total was 0.902 (0.885 for male and 0.915 for female).

Table 1 shows the main characteristics of the study. A total of 1229 Peruvian adolescents participated in the study, of whom 622 were girls and 607 were boys. The CHU9D utility index for the total sample was 0.890. This score was slightly higher in boys (0.887) than in girls (0.867). It was also found that for both sexes, it was statistically significant (*p* < 0.15). The ceiling effect was somewhat higher for males (32.6%) than for females (29.7%). However, a reduction was observed as the age of the adolescents increased with the exception of 17 years. Statistically significant differences were also found in the CHU9D Utility Index between ages as well as between BMI weight groups. 

Table 2 shows the CHU9D dimension levels. The majority of adolescent students reported being in good health according to the CHU9D classification. It is visible that several dimensions had a more diverse range of scores. The highest percentage was observed in the dimension “Worry” (today I do not feel worried, 70.9%) and the dimension “Annoyed” (today I do not feel annoyed, 65.3%). On the other hand, the lowest percentage was in the “Tired” dimension (today I do not feel tired, 42.3%).

Table 3 shows the CHU9D Utility Index of the sample segmented by sex, according to age and BMI category. The boys, aged 13 years, reported a mean of 0.915, which was the highest. In the case of girls, a mean of 0.922 was obtained for the age of 12 years. In contrast, the lowest mean was found for the age of 15 years, both for boys, with a mean of 0.859, and for girls with a mean of 0.814. It can also be observed that the highest ceiling effect was found at the age of 12 years in both sexes, boys (55.6%) and girls (51.5%).

In relation to the weight categories (Table 3), we can see that there is a minimal difference in low weight between both sexes, boys and girls. In terms of normal weight, there was a slight difference between the two sexes. In the overweight category, there was a difference in favor of boys. In the obesity category, there was a difference between both sexes, with boys showing a higher percentage than girls.

In short, there was a statistically significant difference between the sexes for each weight category in the CHU9D Utility

Table 4 shows the distribution of adolescents according to health status. 

We considered showing only the percentages above 0.5 in order to obtain a more representative sample. The total sample of the most common states amounted to 750 adolescents (61.4%), while the least common states had a figure of 479 (38.6%). The table showed how 31.2% of adolescents reported a perfect health status 11111111. The second most common state of health was 111112111, with a total of 98 adolescents. Eight percent of the sample had no problems with worry, sadness, pain, tiredness, discomfort, schoolwork, sleep status, routine, and daily activities.

## 4. Discussion

This research provides relevant information on normative values of the CHU9D in young Peruvian students. The results of this study provide additional support for the validity of the CHU9D for its application in the economic evaluation of health interventions with adolescents aged 12–17 years. Notably, the present article is the first to measure the CHU9D in the Peruvian adolescent population. 

The CHU9D has shown an admissible category of interior robustness, is consistent and reliable in another study conducted in an adolescent population [14]. Along these lines, during this study, the Cronbach’s α score was 0.902, similar to other studies in this population [9], with above 0.70 considered the minimum acceptable [21]. 

The main finding of this study was a good HRQoL in the Peruvian adolescent population, with a score of 0.890 on the CHU9D Utility Index. In this sense, there is scientific evidence from different studies, in which the CHU9D has been applied, where a mean score similar (0.810–0.850) to ours (0.890) was obtained [9,18,22,23]. Along the same lines as our study, several pieces of evidence indicate that HRQoL is perceived as good in the adolescent population [9,18,24]. Regarding the results of the dimensions, the vast majority of the students stated that they were in good health, with high life satisfaction, no disabilities, and no chronic diseases [9,18]. In all studies, it can be observed that there was a good quality of life, which is not considered as a surprising fact due to the type of population recruited.

The CHU9D-DK in this case shows that the highest proportion of students was at the highest level of all dimensions, except for the dimension “tired”, coinciding with the results of our CHU9D study [9]. Similarly, very similar results were found in a study [22] where the CHU9D was also applied in adolescents, who reported excellent health (29.5%), and a small sample reported poor health (0.9%). This finding is again in line with our study, where a total of 31% reported perfect health, while 0.5% reported poor health. 

In addition, a pilot study [25] conducted in China found that 90.3% of the total participants stated that their health status was good, obtaining a mean utility of 0.810, which is similar to our study (0.890). Regarding the dimensions, the highest percentage was obtained in the item: “Today I have no problems with my daily routine” with 86.3%, in contrast to our study whose percentage was found in the dimension of worry “Today I do not feel worried” with 70.9%. The second highest percentage was for both studies in the pain dimension: “I do not have any pain today” with 65.5% compared to our study with 69.8%. The third highest score was for both studies in the sadness dimension: “I do not feel sad today” with 58.8% in contrast to our result with 67.0%. The lowest percentage was also seen for both studies in the dimension of tiredness “Today I do not feel tired” with 19.3% and with 42.3% in our case.

On the other hand, studies were found whose CHU9D utility index differed from the mean obtained in our study [14,23,24]. In the study [24] applied to Australian adolescents, it showed some differences with respect to our research. This study obtained an overall mean of 0.780 compared to our results, which showed a mean of 0.890. The study indicated that the boys’ mean was 0.810, statistically significantly different from the girls’ mean of 0.760. This result does not coincide with our findings, which showed a mean of 0.887 for boys and 0.867 for girls. It was also observed in both studies that younger girls had significantly higher utility values than boys. This trend changed from age 13 to 17, where boys showed higher utility values than girls. Along the same lines, another study was analyzed that showed a difference from our results (0.890), probably due to the large difference in sample size and study characteristics [23]. The discrepancy of our study with the rest of those mentioned may be due to the diversity and context [26] presented in each population where the CHU9D is applied. Our study focused on Peruvian adolescents who presented socio-economic [27], cultural and ethnic characteristics, typical of Latin America [28], unlike from those found in European or Asian studies [29].

Comparisons of the CHU9D with other instruments to assess adolescent HRQoL were also made. In the Australian comparative study [30], a comparison was made between the CHU9D and the PedsQL, where a mean of 0.720 was obtained in contrast to our study (0.890). This contrast may also be due to cultural and socio-economic differences. It was concluded that both instruments performed well in the assessment of HRQoL in this population, confirming their high inter-instrument conformity as well as convergent and known-group validity [30]. The same is the case in another comparative study [31] of the utility of the CHU9D and the HUI2, where the results indicated a moderate level of agreement between the two instruments, indicating that this study supports the validity of the CHU9D instrument in adolescents.

Below is a synoptic table (Table 5) with the characteristics of the selected articles. All these studies report the good quality of life perceived by adolescents through the CHU9D instrument

Having normative data of Peru would be of great interest as it allows us to have information on the health status of the adolescent population, providing access to comparisons of HRQoL in different types of populations with or without pathologies. The aim is to improve HRQoL and, above all, to develop cost-effective treatment and preventive programs that incorporate the needs and priorities of adolescents. Normative data can also be effectively applied to the economic evaluation of interventions aimed at improving quality of life and the development of health programs for this age group, meeting their needs [30]. Along the same lines, the study applied to Australian adolescents also highlights the need to design a health approach that improves the health of young adolescents and is adapted to their preferences [22]. Similarly, in a study using the CHU9D in an adult population in the Netherlands, the aim was to develop a set of Dutch Child Health Utility 9D values, an indicator of quality of life, to be used to create increasing years of quality of life [32].

Finally, the present study has some limitations. First, the survey was conducted online, so only adolescents with Internet access were able to participate. However, Internet use in this population has a practical and recognized way of accessing this resource, unlike in other populations. On the other hand, the nature of our sample was also considered as a limitation, as most of the participants stated in the survey that they were healthy.

## 5. Conclusions

All in all, the findings of this study have shown that adolescents in school have a healthy sense of HRQoL. It was shown that boys reported slightly better health than girls. In addition, HRQoL seems to worsen with age, probably due to the changes adolescents face during this phase of their lives. The study concludes that the CHU9D can be effectively applied to economic evaluations for interventions to improve the quality of life of adolescents. Therefore, further research is considered necessary with the aim of improving the HRQoL of the adolescent population, with larger samples, in different contexts, and including adolescents with different pathologies.

## Figures and Tables

**Table 1 jpm-11-01272-t001:** Sample characteristics. CHU9D adolescents’ population normative values.

			CHU9D Utility Index					Ceiling Effect	
	*n*	(%)	Mean	SD	Median	RI	*p*	*n*	(%)
**Total**	1229	100	0.890	0.165	0.938	0.152		383	31.2
**Gender**									
Female	622	50.6	0.867	0.115	0.920	0.256	0.15 *	185	29.7
Male	607	49.4	0.887	0.125	0.910	0.214		198	32.6
**Age**									
12	200	16.3	0.917	0.118	1	0.125	0.00 ^ŧ^	107	53.5
13	201	16.4	0.907	0.118	0.952	0.141		85	42.3
14	199	16.2	0.883	0.116	0.883	0.216		63	31.7
15	200	16.3	0.837	0.128	0.823	0.246		40	20
16	212	17.2	0.852	0.112	0.861	0.194		37	17.5
17	217	17.7	0.868	0.113	0.868	0.197		51	23.5
**IMC category**									
Low weight	6	5	0.838	0.119	0.870	0.213	0.25 ^ŧ^	1	16.7
Normal weigh	587	47.8	0.871	0.117	0.903	0.235		156	26.6
Overweigh	502	40.8	0.888	0.120	0.948	0.214		180	35.9
Obesity	134	10.9	0.864	0.133	0.920	0.283		46	34.3

* *p* for Mann–Whitney U test; ^ŧ^
*p* for chi-square test.

**Table 2 jpm-11-01272-t002:** Responses to the Child Health Utility 9D.

Dimension/Level	*n*	%
*Worry*		
I do not feel worried today	871	70.9
I feel a little bit worried today	321	26.1
I feel a bit worried today	26	2.1
I feel quite worried today	6	5
I feel very worried today	5	4
*Sad*		
I do not feel sad today	823	67.0
I feel a little bit sad today	364	29.6
I feel a bit sad today	37	3.0
I feel quite sad today	2	2
I feel very sad today	3	2
*Annoyed*		
I do not feel annoyed today	803	65.3
I feel a little bit annoyed today	387	31.5
I feel a bit annoyed today	35	2.8
I feel quite annoyed today	4	3
I feel very annoyed today	0	0
*Tired*		
I do not feel tired today	520	42.3
I feel a little bit tired today	584	47.5
I feel a bit tired today	102	8.3
I feel quite tired today	15	1.2
I feel very tired today	8	7
*Pain*		
I do not feel annoyed today	858	69.8
I feel a little bit annoyed today	317	25.8
I feel a bit annoyed today	51	4.1
I feel quite annoyed today	3	2
I feel very annoyed today	0	0
*Sleep*		
Last night, I had no problems sleeping	813	66.2
Last night, I had a some problems sleeping	359	29.2
Last night, I had a few problems sleeping	47	3.8
Last night, I had many problems sleeping	7	6
Last night, I could not sleep at all	3	2
*Daily*		
I have no problems with my daily routine today	763	62.1
I have a few problems with my daily routine today	448	36.5
I have some problems with my daily routine today	0	0
I have many problems with my daily routine today	13	1.1
I cannot do my daily routine today	5	4
*Schoolwork/homework*		
I have no problems with my schoolwork/homework today	802	65.3
I have a few problems with my schoolwork/homework today	373	30.3
I have some problems with my schoolwork/homework today	42	3.4
I have many problems with my schoolwork/homework today	7	6
I cannot do my schoolwork/homework today	5	4
*Able to join activities*		
I can join in with any activities today	774	63.0
I can join in with most activities today	380	30.9
I can join in with some activities today	55	4.5
I can join in with a few activities today	13	1.1
I can join in with no activities today	7	6

CHU9D: Child Health Utility 9D.

**Table 3 jpm-11-01272-t003:** Study sample characteristics. CHU9D adolescents’ population normative values by gender.

			CHU9D Utility Index									Ceiling effect	
	Male	Female	Male				Female					Male	Female
	*n* (%)	*n* (%)	Mean	SD	Median	RI	Mean	SD	Median	RI	*p* *	*n* (%)	*n* (%)
Age													
12	99 (16.3)	101 (16.2)	0.912	0.124	1	0.204	0.922	0.113	1	0.852	0.895	55 (55.6)	52 (51.5)
13	101 (16.6)	100 (16.1)	0.915	0.110	0.952	0.124	0.898	0.125	0.952	0.194	0.493	45 (44.6)	40 (40)
14	100 (16.3)	100 (16.1)	0.890	0.111	0.920	0.214	0.875	0.122	0.915	0.228	0.645	29 (29.3)	34 (34)
15	100 (16.5)	100 (16.1)	0.859	0.121	0.894	0.240	0.814	0.131	0.785	0.247	0.260	23 (23.0)	17 (17)
16	100 (16.5)	112 (18.0)	0.867	0.110	0.897	0.192	0.839	0.112	0.845	0.199	0.560	20 (20.0)	17 (15.2)
17	108 (17.8)	109 (17.5)	0.879	0.107	0.915	0.172	0.857	0.117	0.876	0.237	0.235	26 (24.1)	25 (22.9)
**IMC category**													
Low weight	4 (7)	2 (3)	0.859	0.133	0.879	0.247	0.794	0.111	0.794	0.716	0.800	1 (25)	1 (50)
Normal weigh	289 (47.6)	298 (47.9)	0.880	0.111	0.915	0.215	0.861	0.121	0.882	0.256	0.970	81 (28)	75 (25.2)
Overweigh	235 (38.7)	267 (42.9)	0.898	0.115	0.952	0.188	0.879	0.125	0.920	0.224	0.128	88 (37.4)	92 (34.5)
Obesity	79 (13.0)	55 (8.8)	0.878	0.129	0.952	0.272	0.843	0.138	0.812	0.286	0.187	28 (45.4)	18 (32.7)

* *p* for Mann–Whitney U test.

**Table 4 jpm-11-01272-t004:** Distribution of health.

CHU9D	Frequency	Valid Percentage	Accumulative Percentage
Health Status			
111111111	383	31.2	31.2
111112111	98	8.0	39.2
222222222	76	6.2	45.4
221222222	33	2.7	48.1
112112111	32	2.6	50.7
111112211	28	2.3	53
222222221	12	1.0	54
111111112	10	0.8	54.8
112112211	9	0.7	55.5
111112121	8	0.7	56.2
222212111	8	0.7	56.9
222223222	8	0.7	57.6
111113111	7	0.6	58.2
111122111	7	0.6	58.8
211112111	7	0.6	59.4
111122211	6	0.5	59.9
112122111	6	0.5	60.4
211111111	6	0.5	60.9
222213111	6	0.5	61.4
Most Common States	750	61.4	61.4
Other Status *	479	38.6	100
Total	1229	100	100

* It was considered “Other Status” for *n* < 5.

**Table 5 jpm-11-01272-t005:** Studies reporting on adolescents’ perceived good quality of life using the CHU9D instrument.

Year	Study Reference	Country	Age Category	*n*	Utility
2021	Present study	Perú	12–17	1229	0.890
2021	Le, Richards-Jones et al., 2021	Australia	11–17	2967	0.780
2015	Furber, Segal et al., 2015	Australia	5–17	590	0.739
2011	Ratcliffe, Couzner et al., 2011	Australia	11–13	45	0.850
2019	Petersen, Ratcliffe et al., 2019	Denmark	11–26	272	0.840
2016	Ratcliffe, Huynh et al., 2016	Australia	11–17	1892	-
2014	Xu, Chen et al., 2014	China	9–17	815	0.810
2020	Qi, Qin et al., 2020	China	10–13	4388	0.780
2018	Yang, Chen et al., 2018	China	13–17	823	0.810
2018	Petersen, Chen et al., 2018	Australia	15–17	775	0.720
2012	Ratcliffe, Stevens et al., 2012	Australia	11–17	216	0.844

## Data Availability

The datasets used during the current study are available from the corresponding author on reasonable request.

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
