# Peer review of "CHU9D Normative Data in Peruvian Adolescents"

_jpm, 2021, doi:10.3390/jpm11121272_

Round 1
Reviewer 1 Report
I have read with great interest the manuscript entitled "Chu9d Normative Data In Peruvian Adolescents".
The authors must answer the following questions in order to consider their work :
1) How were participants of Peruvian origin selected?
2) Why was the study conducted with Peruvian adolescents?
3) The authors do not justify why they have analysed Peruvian adolescents and how these adolescents were selected.
4) How was the sample size calculated?
5) In Sample Size the authors indicate that the sample consisted of 622 females and 607 males, but the percentages they indicate are not correct.
6) Why were only participants over 12 years of age considered? The CHU9D questionnaire is validated for patients older than 11 years, why were 11-year-old participants not included?
7) The authors do not indicate when the data collection took place.
8) The bibliographic references are not described according to the journal's guidelines.
Author Response
Dear Reviewer,
Thank you for your review of our manuscript. We have carefully considered your comments and believe that the quality of the paper has improved after incorporating your suggestions. Below are our responses to your suggestions:
- How were participants of Peruvian origin selected?
Authors' response: Adolescence is a period when various changes occur in the organism and personality is being formed, and these changes have an influence on HRQOL, so it is very important to have information on HRQOL at this age.
As to why Peru, HRQOL in adolescence is different in each country since it is influenced by cultural, political and economic factors, among others, and therefore specific data should be available for each country. In this sense, to the best of our knowledge, there are no reference data on Peruvian adolescents, so this study contributes to provide the scientific community with data on HRQOL that are not currently available.
Furthermore, one of the questions that formed part of the sociodemographic questionnaire included whether they had Peruvian nationality; therefore, as it can be considered one of the inclusion criteria, we have proceeded to include it in the text.
- Why was the study conducted with Peruvian adolescents?
Authors' response: In a systematic literature search on health-related quality of life (HRQoL) in South American countries, and more specifically in Peru, we found that there is a paucity of published data on HRQoL focused on the adolescent population group.
In order to highlight the importance of this study, the following paragraph was inserted in the introduction: "Despite the increase in HRQoL studies, there are few studies aimed at the Latin American population. To our knowledge there is no study in which the CHU9D instrument has been used in adolescents in Peru".
- The authors do not justify why they have analysed Peruvian adolescents and how these adolescents were selected.
Authors' response: The justification for analyzing Peruvian adolescents has been included in the answer to the previous comment. As mentioned in the previous comment in response to the first of your questions, we have proceeded to insert the above information.
All the adolescents self-reported that they had Peruvian nationality. Thanks to your suggestion, it has been included as an explicit inclusion criterion that the participant should have Peruvian nationality.
This information has been included and therefore the edition of the "participants" section. If you consider that more information should be added, please let us know.
- How was the sample size calculated?
Authors' response: A random sample of 1153 individuals is sufficient to estimate, with 95% confidence and a precision of +/- 2.5 percentage units, a population percentage that is expected to be around 25%.
- In Sample Size, the authors indicate that the sample consisted of 622 females and 607 males, but the percentages they indicate are not correct.
Authors' response: Thank you for your comment, it was an error in the data transcription. The percentage of 622 women has been corrected, so the percentage for them is 50.6, and the percentage for 607 male participants is 49.4 as shown in table 1.
- Why were only participants over 12 years of age considered? The CHU9D questionnaire is validated for patients older than 11 years, why were 11-year-old participants not included.
Authors' response: As indicated in the methodology section, the study was conducted in secondary schools in Peru, although there are some students who are 11 years old when they enter secondary school, it was decided to exclude them since 11 years old marks the transition between primary and secondary school and these data would only be representative of those students who are 11 years old and were born in the last quarter of the year, so that not all those born throughout the year would have been included. On the other hand, not to extend the study to 11-year-olds for reasons of difficulty.
- The authors do not indicate when the data collection took place.
Authors' response: Data collection was carried out from September to December.
We appreciate your comments and have included the dates on which the study was conducted.
- The bibliographic references are not described according to the journal's guidelines
Authors' response: In response to your suggestions, the following modifications have been made.

Reviewer 2 Report
I believe that the article is original and pertinent and of relevance for research in the Latino population. overall the article is well structured and easy to understand. My observations are intended to contribute to the improvement of the manuscript in the following sense.
Although it seems to have coherence in the research in general and in the application of the method, it is necessary to document more about:
How the sample size was established.
There are statistical tests that were applied, but no data is presented or discussed.
It is necessary to document the results of the internal consistency of the investigation
It is convenient to include for clinical measurements how sensitivity and specificity were addressed; it is not clear how it was measured and which variables were controlled for.
Round 2
Reviewer 1 Report
The authors have revised manuscript according to the review opinions, so I think it can be published in its current state.
Author Response
Dear editor, we have listened to your suggestions and have restructured and added the information as you indicated. We also apologise for having made a mistake in including Cronbach's alpha and have already modified this value in the results and included it in the discussion. In addition, we will proceed to attach the database in response to the email that is automatically generated when uploading the minor revisions, as it is not possible to attach this document here. This way, you can check the value yourself.
Thank you very much for your suggestions.